# Experimental realisations of the fractional Schrödinger equation in the temporal domain

Shilong Liu [1,2] ✉, Yingwen Zhang [1,3], Boris A. Malomed[4,5] & Ebrahim Karimi [1,3] ✉

The fractional Schrödinger equation (FSE)—a natural extension of the standard Schrödinger equation—is the basis of fractional quantum mechanics. It can be obtained by replacing the kinetic-energy operator with a fractional derivative. Here, we report the experimental realisation of an optical FSE for femtosecond laser pulses in the temporal domain. Programmable holograms and the single-shot measurement technique are respectively used to emulate a Lévy wave-guide and to reconstruct the amplitude and phase of the pulses. Varying the Lévy index of the FSE and the initial pulse, the temporal dynamics is observed in diverse forms, including solitary, splitting and merging pulses, double Airy modes, and "rain-like" multi-pulse patterns. Furthermore, the transmission of input pulses carrying a fractional phase exhibits a "fractional-phase protection" effect through a regular (non-fractional) material. The experimentally generated fractional time-domain pulses offer the potential for designing optical signal-processing schemes.

Fractional Schrödinger Equation (FSE) was originally introduced, as a quantum-mechanical model, by means of the Feynman-integral formalism for particles moving by Lévy flights[1,2]. This means that the average distance $L$ of the randomly walking classical particle from its initial position grows with time $t$ as

$$L \sim t^{1/\alpha},\tag{1}$$

where $\alpha$ is the Lévy index (LI)[3]. In the case of $\alpha = 2$, it is the usual random-walk law for a Brownian particle. The Lévy-flight regime, corresponding to $\alpha < 2$, implies that the diffusive walk is faster than Brownian. At the classical level, it is performed by random leaps. In the corresponding FSE, which is the basis of the fractional quantum mechanics[1], the kinetic-energy operator has the form of $\left(-\nabla^2\right)^{\alpha/2}$ which implies the same relation between the length and time scales in solutions of the FSE as indicated by Eq. (1)—here $\nabla^2$ is the Laplacian. In the context of fractional quantum mechanics, several setups have been elaborated, based on various potentials included in the FSE. These are, in particular, the fractional generalizations of the Bohr atom and harmonic oscillator[4,5].

Experimentally, the main challenge in implementing the fractional quantum phenomenology is to find a physical setting featuring Lévy flights with arbitrary LI, which would impart a fractional-phase shift to the particle's wave function, and thus realize the FSE. An alternative is to use a condensed-matter setting to realize the spatial-domain FSE[6]. In this case, the key ingredient is a 1D Lévy crystal with LI taking values $1 < \alpha \leq 2$, which is introduced in the form of a 1D infinite-range tight-binding chain[7,8]. However, building a Lévy crystal with an arbitrary LI in the solid-state system remains challenging.

An alternative, proposed by Longhi in 2015[9], is to use the similarity of the quantum-mechanical Schrödinger equation to the paraxial wave-propagation equation in optics. The proposed protocol employed transverse light dynamics in aspherical optical

[1]Department of Physics, University of Ottawa, 25 Templeton, Ottawa, ON K1N 6N5, Canada. [2]State Key Laboratory of Modern Optical Instrumentation, College of Optical Science and Engineering, Zhejiang University, 310027 Hangzhou, Zhejiang, China. [3]National Research Council, 100 Sussex Dr, Ottawa, ON K1A 0R6, Canada. [4]Department of Physical Electronics, Faculty of Engineering, and Center for Light-Matter Interaction, Tel Aviv University, Tel Aviv 69978, Israel. [5]Instituto de Alta Investigación, Universidad de Tarapacá, Casilla 7D, Arica, Chile. ✉e-mail: dr.shilongliu@gmail.com; ekarimi@uottawa.ca

cavities designed in the 4*f* configuration. Under the paraxial approximation and ignoring losses originating from optical elements and diffraction, the output transverse modes correspond to eigenfunctions of the FSE with a chosen value of LI. However, in such complex optical setups, loss is a critical problem, which requires one to introduce compensating gain[10], thus making the analysis more cumbersome. Due to these problems, experimental implementation of the FSE has not yet been achieved.

Most of the current studies dealing with FSE were focused on simulations. In particular, the light-beam propagation in these models gives rise to a breather-like behavior[11,12]. Another topic of great interest is the nonlinear FSE, which includes the Kerr nonlinearity of the optical material. It produces diverse families of optical fractional solitons, such as gap solitons[13], multipoles[14], trapped modes[15], Airy waves[16], as well as solitary vortices[17], see recent review[12]. Setups supporting fractional solitons offer various applications for the design of photonic data-processing schemes[18–20]. Further, by combining the FSE and a complex potential, the parity-time symmetry and its breaking may be featured[21–25].

Here, we report one experimental realization of an optical system representing the FSE in the temporal domain. The main principle is to transform the temporal FSE into the frequency domain and apply a spectral phase shift representing a Lévy waveguide with an arbitrary value of LI, $0 \leq \alpha \leq 2$. Employing a recently developed single-shot measurement technique—in particular, in the form of spatial-spectrum interferometry (SSI)[26]—we perform the pulse reconstruction based on a single measurement. In the experiment, we used two holograms in the pulse-shaper system. The first one morphs an appropriate phase pattern in input pulses, while the second hologram acts as the optical Lévy waveguide, with particular values of coefficient $D$ of the fractional group-velocity-dispersion (GVD) and LI $\alpha$. Passing the second hologram, the pulse receives the propagation-phase shift emulating the passage through a fractional dispersive waveguide. Thus, the temporal FSE is realized in this work in an indirect form. The creation of a fractional dispersive material and direct realization of the FSE is still an open challenge[12].

Thus, based on the initial conditions and propagation parameters, three main scenarios of the temporal dynamics are explored in this work. First, we launch an initial femtosecond pulse with a second-order spectral phase, representing the frequency chirp[27]. Several outcomes are evidenced by observing solitary pulses, double Airy modes, "rain-like" multi-pulse trains, and collisions between pulses. To characterize the outcomes, we reconstruct their Wigner function in the time–frequency chronocyclic space[28], with the interference fringes exhibiting a sub-Fourier structure. In terms of quantum mechanics, it represents sub-Planck features, which correspond to values of action $< \hbar/2$ (i.e., smaller than the minimum action fixed by the Heisenberg's uncertainty principle[29]). Thus, the sub-Planck phenomena may be emulated using the wave propagation in optics[30–32]. Such small action shifts are essential in quantum mechanics, as they are sufficient to make initially identical coherent states distinguishable. In optics per se, the corresponding sub-Fourier features can be used for high-precision measurements, somewhat similar to the technique based on super-oscillations[33–35].

Next, we launch the input pulse carrying the third-order spectral phase, which naturally leads to an Airy wave evolution in the temporal domain. Our results show that the trajectory of the leading lobe of Airy pulses can be effectively controlled by LI. Finally, a "fractional-phase protection" effect is observed when the input pulse with the fractional-phase structure propagates in a regular dispersive waveguide. These results demonstrate remarkable behaviors governed by the FSE, which are relevant as optically realization of the fractional wave dynamics, and may also offer applications to ultrafast signal processing.

## Results
### Realizing the FSE in the temporal domain

When an optical pulse travels in a complex dispersive material, a generalized model for the slowly varying amplitude $\Psi(\tau, z)$ of the electric field may be derived, in the temporal domain, in the known form[1,9,36]:

$$i\frac{\partial \Psi}{\partial z} = \left[ \frac{D}{2}\left(-\frac{\partial^2}{\partial \tau^2}\right)^{\alpha/2} - \sum_{k=2,3\dots} \frac{\beta_k}{k!}\left(i\frac{\partial}{\partial \tau}\right)^k + V(\tau) \right]\Psi, \quad (2)$$

On the right-hand side of Eq. (2), the first term is the fractional time derivative with the LI $\alpha$, where $D$ is the fractional dispersion coefficient. The second term represents the integer derivative that corresponds to the $k$th regular GVD with the coefficient $\beta_k$, and the last term is potential $V(\tau)$. In optics, such a complex dispersion material involving both these ingredients may be realized by means of an artificial photonic structure[37]. In that case, the model defined in Eq. (2) may also include dissipation, i.e., dispersive losses, which are not considered in this work. In the case of $D = 0$ and $V(\tau) = 0$, Eq. (2) reduces to the commonly known propagation equation for linear dispersive media[27]. The fractional Riesz derivative in Eq. (2) is represented by the integral operator[38],

$$\left(-\frac{\partial^2}{\partial \tau^2}\right)^{\alpha/2}\Psi = \frac{1}{2\pi}\int_{-\infty}^{+\infty}\int_{-\infty}^{+\infty} d\theta\, d\omega\, |\omega|^\alpha\, e^{-i\omega(\theta-\tau)}\,\Psi(\theta). \quad (3)$$

We set the external potential to be zero, i.e., $V(\tau) = 0$. Applying the Fourier transform to Eq. (2), it yields a straightforward solution, $\hat{\Psi}(\omega, z)$, in the frequency domain:

$$\hat{\Psi}(\omega, z) = \exp\left[-i\left(\frac{D}{2}|\omega|^\alpha - \sum_{k=2,3,\dots} \frac{\beta_k}{k!}\omega^k\right)z\right]\hat{\Psi}_{\text{input}}(\omega), \quad (4)$$

where $\hat{\Psi}_{\text{input}}(\omega)$ is the respective input profile in the frequency domain. The input $\hat{\Psi}_{\text{input}}(\omega)$, which appears in Eq. (4), may be prepared with the help of a linear pulse shaper[39,40]. Equation (4) suggests an experimentally feasible way to engineer the relevant fractional phase in the frequency domain and thus create a temporal profile corresponding to the solution of the FSE equation.

Figure 1a shows the architecture used to experimentally realize the FSE in the temporal domain. The setup includes three main ingredients: (1) the input ultrafast pulse is morphed by the hologram, as it is done by the traditional pulse shaper; (2) the second hologram adds to the pulse the spectral phase shift emulating the passage through the optical Lévy waveguide; (3) the single-shot SSI technique is used to reconstruct the pulse's amplitude and phase. This regime offers an effective platform to realize and study the pulse dynamics governed by FSE. Further details of the experimental setup are given in Methods and Supplementary Notes 1–3.

The pulse evolution depends not only on the LI, but also on input $\Psi_{\text{input}}(\tau, z = 0)$, see Eq. (4). In particular, the second-order GVD, acting in the shaping segment of length $L_{\text{GVD}}$, produces a phase shift $\phi_{\text{GVD}} = -\beta_2 \times L_{\text{GVD}} \times \omega^2/2$, which corresponds to the input

$$\Psi_{\text{input}}(\tau) = \mathcal{F}^{-1}[\bar{\Psi}_{z=0}(\omega) \cdot \exp(-i\beta_2 L_{\text{GVD}}\omega^2/2)]. \quad (5)$$

Here, $\mathcal{F}^{-1}$ stands for the inverse Fourier transform, and $\bar{\Psi}_{z=0}(\omega)$ is the spectral form of the input from the laser source with the nearly flat phase (low chirp), which has the Gaussian profile (see "The treatment of the FSE" in "Methods"). For the spectral phase $\phi_{\text{GVD}}$, it can be positive or negative depending on the sign of $\beta_2$. For the definiteness's sake, we set $\beta_k$ to be a constant, i.e., $\beta_2 = -21 \times 10^{-3}$ ps²/m, and only discuss effects of $L_{\text{GVD}}$. Thus, we define four quadrants Q1–Q4 in the parameter plane of $(L_{\text{GVD}}, \alpha)$, as shown in Fig. 1b. Areas Q1 and Q2 correspond to the cases with $L_{\text{GVD}} > 0$ and $1 \leq \alpha \leq 2$ or $0 \leq \alpha \leq 1$, respectively, while Q3 and Q4 are defined similarly, but with $L_{\text{GVD}} < 0$. When $\alpha$ is close to 2 (in the B2 area in Fig. 1b), the pulse propagation is

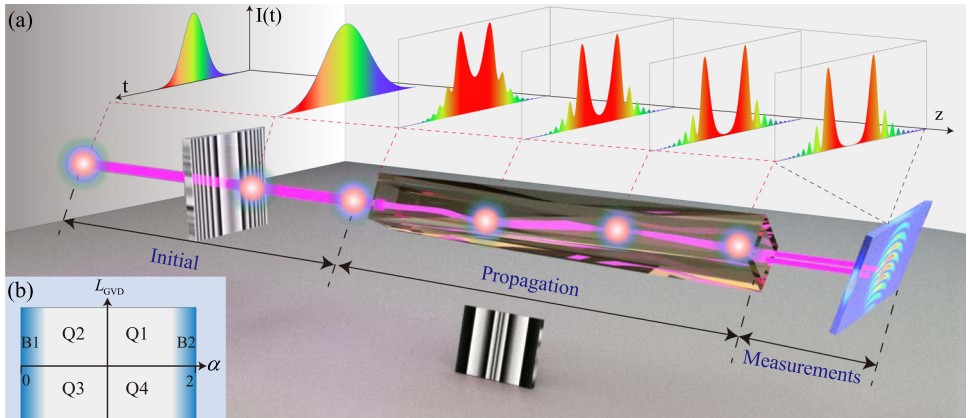

**Fig. 1 | The setup used for the realization of the fractional Schrödinger equation. a** In the initial section, the hologram is installed to shape the input pulse. In the propagation section, the hologram inducing the spectral phase shift, which represents the effect of the fractional GVD, is used to emulate the propagation through the optical Lévy waveguide (for clarity's sake, this hologram is drawn in the rotated form, facing the figure's plane). For the measurements, the single-shot spatial-spectrum interferometry technique is used to measure the pulse's amplitude and phase. The temporal profile of the input, and the evolution of the structure of the propagating beam in the case of Q3, as defined in (**b**), are schematically shown at the top of the figure. **b** Four different quadrants, Q1 to Q4, in the parameter plane of ($L_{GVD}$, $\alpha$). Quadrants Q1 and Q2 correspond to the cases with $L_{GVD} > 0$ and $1 \leq \alpha \leq 2$ and $0 \leq \alpha \leq 1$, respectively. Q3 and Q4 are defined similarly but with $L_{GVD} < 0$. Areas B1 and B2 in (**b**) designate two extreme cases with $\alpha$ close to 0 and 2, respectively.

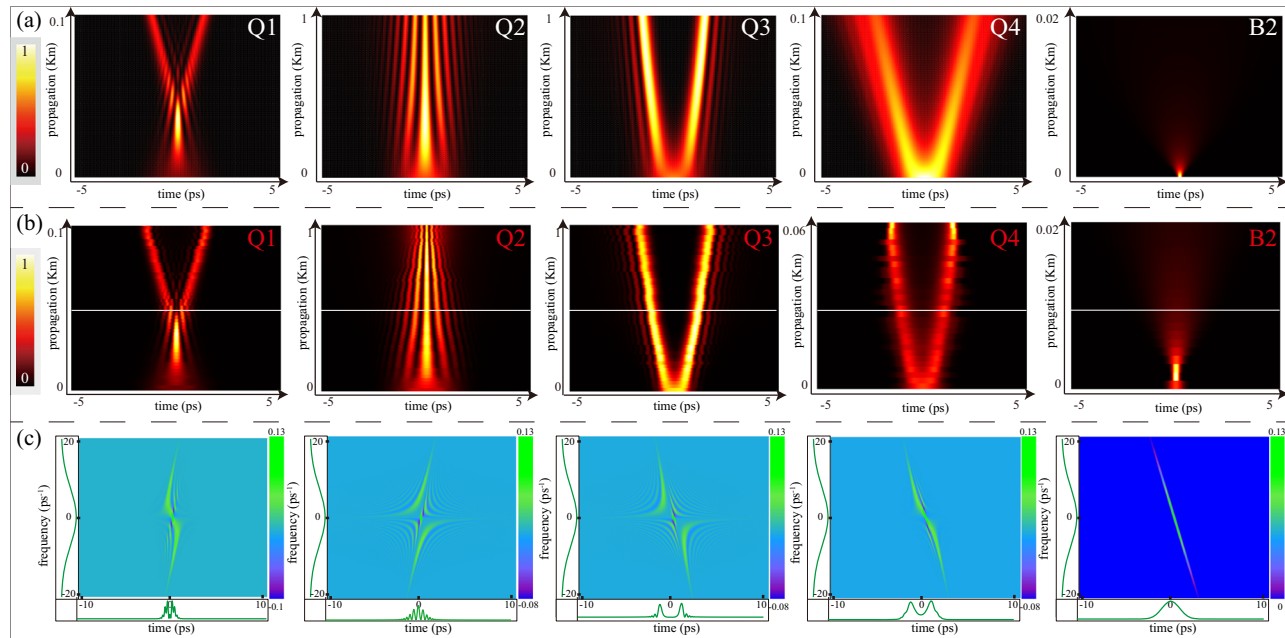

**Fig. 2 | The pulse propagation in the optical Lévy waveguide. a** Simulations of the FSE for different values of LI $\alpha$ and dispersion length $L_{GVD}$, selected as per Eq. (6) for cases Q1 to Q4 and B2 (as defined in Fig. 1b), respectively. Note that time varies in all panels belonging to rows (**a**) and (**b**) from −5 to +5 ps, while the largest values of distance $L_{GVD}$ are different in particular panels. The fractional and regular second-order GVD coefficients in Eq. (2) are fixed as $D = 21 \times 10^{-3}$ ps$^\alpha$/m and $\beta_2 = -21 \times 10^{-3}$ ps$^2$/m, respectively. **b** Experimental results for cases Q1–Q4 and B2. **c** The Wigner functions in the time–frequency space, plotted along the white horizontal lines marked in all panels Q1 to Q4 and B2 in row (**b**). In panels belonging to row (**c**), time varies between −10 and +10 ps, while the frequency varies between −20 and +20 ps$^{-1}$. The color-coding in these panels represents the relative intensity, which is normalized to take values uniformly from minimum to maximum.

close to that in the regular dispersive material[27,41–43]. The effect of the fractional GVD on the pulse propagation is expected to be weak in area B1 in Fig. 1b, which corresponds to small values of LI.

**The case of the second-order spectral phase in the input: the temporal dynamics of femtosecond pulses in the optical Lévy waveguide**

First, the results of simulations of the model for parameters belonging to areas Q1 to Q4, which are defined in Fig. 1b, are presented in Fig. 2a (details of the numerical simulations are given in "Methods"). We fix the value of the second-order GVD coefficient $\beta_2 = -21 \times 10^{-3}$ ps$^2$/m, a

typical value for the single-mode fiber, and set the fractional-GVD coefficient as $D = 21 \times 10^{-3}$ ps$^\alpha$/m, for adequate comparison between the regular and fractional GVD. Values of the parameters of the control set are chosen as,

$$(L_{GVD}, \alpha) = (5, 1.25), (5, 0.25), (-5, 0.25), (-5, 1.25), \quad (6)$$

for Q1 to Q4, respectively. It is seen in Fig. 2a that the model of the optical Lévy waveguide gives rise to diverse dynamical regimes for the temporal pulses. In particular, the case of Q1 shows the splitting of the input pulse into two secondary ones; Q2 demonstrates the generation

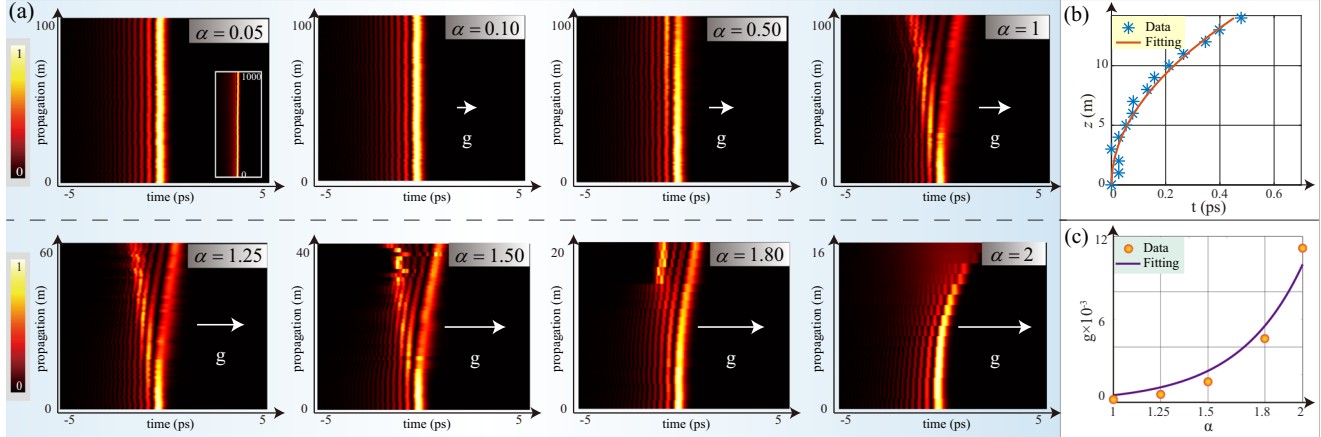

**Fig. 3 | The propagation of self-accelerating Airy pulses in the optical Lévy waveguide. a** shows the beam propagation with different values of LI $\alpha$. The inset in the first panel shows the long-range propagation distance, from $z = 0$ to 1000 m. White arrows designate effective gravity, whose strength is proportional to the length of the arrows. In each panel, the vertical axis represents time extending from −5 to 5 ps; the horizontal axis represents the propagation distance, $z$; the color provides a map of the normalized temporal intensity. **b** The data fit for the accelerating motion of the main lobe of the Airy-shaped pulse for $\alpha = 1.80$. In this plot, the horizontal axis is the distance (in meters), and the vertical one represents time values (from 0 to 0.7 ps). The solid line is the fitting curve defined as per Eq. (9). **c** The experimental data for the virtual gravity $g$, see Eqs. (9) and (11) (orange balls), and its fit (the solid purple line) representing the exponential expression given by Eq. (13).

of multiple pulses, similar to the "soliton rain" structure[44]; in Q3 and Q4, the double profiles may be construed as Airy waves, with pulse broadening observed in the latter case. As concerns, the Airy-shaped beams, the appearance of similar solutions of the spatial-domain FSE was reported by Longhi[9]. Region B2 in Fig. 1b, designated as a reference, corresponds to the regular case with $\alpha = 2$ and $L_{GVD} = 0$.

The respective experimental results, which are the most essential finding reported in the present work, are displayed in Fig. 2b. In all the cases, they closely resemble the predictions of the numerical simulations, cf. Fig. 2a. Note that, looking at the case of Q1 in the backward direction (this is relevant, as the evolution in $z$, governed by Eq. (2), is reversible), one observes the merger of colliding pulses. In Q2, a "soliton rain"-like pulse appears for $z = 0$–1000 m. In this case, the individual pulses cannot be fully resolved because of the limited temporal resolution of the D-shaper (see Supplementary Note 2). Dual pulses are observed in areas Q3 and Q4, where Q3 produces an Airy-like pulse, while Q4 shows two broadened pulses. Area B2 in Fig. 2b shows the experimental results for the regular case of the broadened pulse with $\alpha = 2$. Here, the minimum of the pulse duration is shifted slightly from the expected zero point, which implies a minor difference in the frequency chirp between the reference and signal pulse. This additional chirp mainly comes from the various fused silica optical elements, which could be compensated by adding a small opposite spectral phase on the hologram (as highlighted in Supplementary Fig. 1). In particular, when the input has no frequency chirp (i.e., $L_{GVD} = 0$), the input pulse will split in two, the splitting angle increasing as LI increases from 0 to 2. This special case is considered in the Supporting Material (see Supplementary Movie 1) and Methods.

To fully characterize the pulses in both the temporal and spectral domains, we define the Wigner function[30,31,45],

$$W(t, \omega) = \int_{-\infty}^{+\infty} E\left(t + \frac{t'}{2}\right) E^*\left(t - \frac{t'}{2}\right) e^{i\omega t'} \, dt', \qquad (7)$$

for electric field $E(t)$, at values of $z$ corresponding to the mid positions in panels Q1–Q4 and B2 of Fig. 2b, which are marked by white horizontal lines. It can be concluded from the comparison of panels corresponding to cases Q1 and Q4, and, separately, Q2 and Q3 in Fig. 2c (these pairs of panels are plotted for equal values of LI but opposite values of $L_{GVD}$) that the variation of the GVD leads to rotation of the Wigner-function

distribution in the time–frequency plane. Further, comparing the panels corresponding to Q1 and Q2, and, separately, to Q3 and Q4, demonstrates that LI strongly affects the Wigner-function structure. Namely, smaller LI stretches the interference pattern in the temporal direction. The Wigner function for the reference case of B2 features a smooth rotating distribution without any interference fringes.

### The case of the third-order spectral phase in the input: control of the curvature of Airy pulses by the Lévy index

In this section, we address the evolution of the input pulse carrying the third-order spectral phase. To realize it, an initial temporal profile is taken as

$$\Psi_{\text{input}}(\tau) = \mathcal{F}^{-1}[\tilde{\Psi}_{z=0}(\omega) \cdot \exp(-i\beta_3 L_{GVD}\omega^3/6)], \qquad (8)$$

with $\beta_3 = 0.1 \times 10^{-3} \text{ps}^3/\text{m}$ and $L_{GVD} = 500$ m. This input, which is produced by the pulse shaper, cf. Eq. (5), generates a propagating pulse in the form of the Airy function[46–49]. The experimentally observed evolution is displayed in Fig. 3. We first set $\alpha = 0.05$ and observe the pulse shaped as the Airy function, which keeps its profile during the propagation governed by the FSE-emulating setup. Note that the trajectory of the Airy pulse does not accelerate (the inset in the right panel of Fig. 3a displays the propagation pattern up to $z = 1000$ m). With the increase of $\alpha$, a new accelerating branch of the Airy structure originates from the input. The acceleration increases with the growth of $\alpha$, and the propagating pulse recovers the usual Airy shape at $\alpha = 2$, the same as produced by the regular Schrödinger equation[47–49].

In the framework of the latter equation, the accelerating trajectory of the Airy beam (a bending one, in the other rendition), can be predicted by the model of the classical ballistic motion of a particle with mass $M$[50]. Here, the trajectory can be modeled in terms of motion in a virtual gravity field $g$, which is directed along the time axis $t$ (see the arrows depicted in Fig. 3). Then, the wave packet moves like a particle in the plane of $(t, z)$, starting from the origin, $(0, 0)$:

$$t = \frac{g}{2M}z^2 \qquad (9)$$

(here, we set $M = 1$). For the sake of clarity, we here focus on the main lobe of an Airy beam in the plane of $(t, z)$.

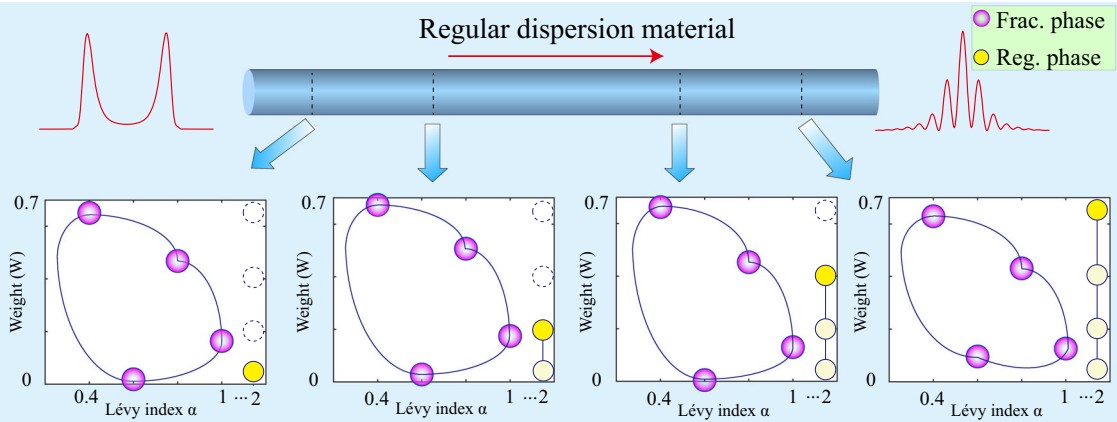

**Fig. 4 | The demonstration of the "fractional-phase protection".** Here, the central bar represents a waveguide with nonfractional GVD, and the left plot is the initial intensity profile with the fractional phase, as produced by the shaper with ($\alpha = 1.20$, $L_{\text{frac}} = 70$ m), and the right plot shows the corresponding output profile, affected by the action of the second-order GVD in the course of the propagation. The four bottom panels show the weightings and LI for the reconstructed phase structure given by Eq. (14) for propagation distances $z = 1$, 3, 6, and 9 m, respectively. In these panels, the four purple balls represent four different fractional-phase ("Frac. phase") terms in the first line of Eq. (14), which are generated by input-phase terms with parameters $\left(\alpha_j, W_j\right) = (0.4, 0.73)$, (0.6, 0.07), (0.8, 0.51), and (1.0, 0.22), respectively. The horizontal axis in each panel represents the set of values of LI $\alpha_j = 0.4$, 0.6, 0.8, 1, and 2, respectively; the vertical axis represents values of the relative weight $W_j$ from 0 to 0.7. Yellow balls denote the relative weights of the second-order regular phase ("Reg. phase") term $W_{k=2}$. The solid line, connecting the purple or yellow ball, is used to show their relative distribution.

Comparing Eq. (9) to the experimental results presented in Fig. 3a, we conclude that values of the effective gravity corresponding to

$$\alpha \leq 0.5, \quad \text{and} \quad \alpha = 1, 1.25, 1.5, 1.8, 2 \tag{10}$$

are, respectively,

$$g_{\text{exper}} = 0, \text{ and } g_{\text{exper}} = [0.21, 0.58, 1.49, 4.61, 11.15] \times 10^{-3}. \tag{11}$$

The corresponding values of $g$ extracted from simulations (see Supplementary Movie 2) are

$$g_{\text{theory}} \leq 10^{-4}, \text{ and } g_{\text{theory}} = [0.17, 0.45, 1.11, 4.04, 8.90] \times 10^{-3} \tag{12}$$

The comparison of Eqs. (11) and (12) demonstrates that the crude mechanical model, based on Eq. (9), provides a reasonable approximation. Further, Fig. 3b provides a fit of the experimental data for the main lobe of the propagating Airy-shaped pulse to Eq. (9), at $\alpha = 1.80$. According to experimental data, values of the adequate acceleration, in the entire interval of $0 \leq \alpha \leq 2$, obey an empirical relation,

$$g \approx p_1 \exp(p_2 \alpha), \tag{13}$$

where $p_{1,2} > 0$ are fitting parameters. The exponential dependence on the LI in Eq. (13) is explained by the transition from the integral operator Eq. (3) to the regular local derivative, as the LI is approaching the usual limit value, $\alpha = 2$. Figure 3c shows fitting of the relation between $g_{\text{exper}}$ and $\alpha$ to the exponential dependence defined in Eq. (13). The fitting corresponds to $p_1 = 2.748 \times 10^{-5}$ and $p_2 = 2.947$, respectively.

**The case of the fractional-order spectral phase in the input: the "fractional-phase protection" effect**

While building an actual optical Lévy waveguide that can support the pulse propagation is a challenge, the use of the fractional phase may provide an advantage in some other cases. In particular, the input pulse, $\hat{\Psi}(z = 0)$, containing a sum of fractional-phase terms with different LI values $\alpha_j$ and different weights, will acquire a nonfractional spectral phase shift in a regular dispersion material, according to the general relations,

$$\hat{\Psi}(z = 0) = |\hat{\Psi}(z = 0)| \exp\left[-i \sum_j W_j \cdot |\omega|^{\alpha_j}\right]$$

$$\hat{\Psi}(z = L) = \hat{\Psi}(z = 0) \exp\left[i \sum_{k=2,3,\dots} W_k \cdot \omega^k L\right], \tag{14}$$

where $\hat{\Psi}(z = 0)$ and $\hat{\Psi}(z = L)$ are the input and output, respectively. $W_j = D L_{\text{frac},j}/2$ represents the weight of fractional phase on each $\alpha$, and $W_k = \beta_k/k!$ gives the corresponding contribution on the regular $k$th dispersion. Thus, the fractional part in the input phase is kept unaltered in the transmission, according to Eq. (14), which demonstrates an effect of the "fractional-phase protection". It may be useful for the design of optical data-transmission schemes.

Figure 4 presents the results of testing this effect. To this end, the initial pulse, carrying the fractional phase, is injected into the regular (nonfractional) dispersive segment, which is represented by the second hologram in the present setup. As a result, regular and fractional-phase shifts are mixed in the output pulse. Even if the regular phase change may be very complex, as shown by Eq. (14), the fractional parts are still kept. To demonstrate this, we select four initial pulses with parameters $\left(\alpha_j, W_j\right) = (0.4, 0.73)$, (0.6, 0.07), (0.8, 0.51), and (1.0, 0.22), respectively, and the propagation distance $z$ varying from 0 to 10 m. We present the scheme and reconstruct the so-produced spectral phase in Fig. 4. The results confirm the stability of the fractional phase. Four panels show, using purple balls, weights of four terms in the reconstructed phase, corresponding to four particular values of $\alpha < 2$, after passing distances $z = 1$ m, 3 m, 6 m, and 9 m. For comparison, the yellow balls represent the regular (nonfractional) part of the phase, corresponding to $\alpha = 2$. The height of the yellow mark increases during the propagation, while the fractional terms remain mostly constant, with the corresponding marks forming a closed contour, shown by blue lines. These results directly confirm the validity of the "fractional-phase protection" effect. Full results of the experiments and simulations represented in Fig. 4 are shown in Supplementary Movie 3. It should be noted the "fractional-phase protection" effect may not hold for all values of $\alpha$. With $\alpha$ becoming closer to 2, the profile of the fractional spectral phase will become more similar to the nonfractional case of $k = 2$. When $\alpha = 2$, the two types of phases

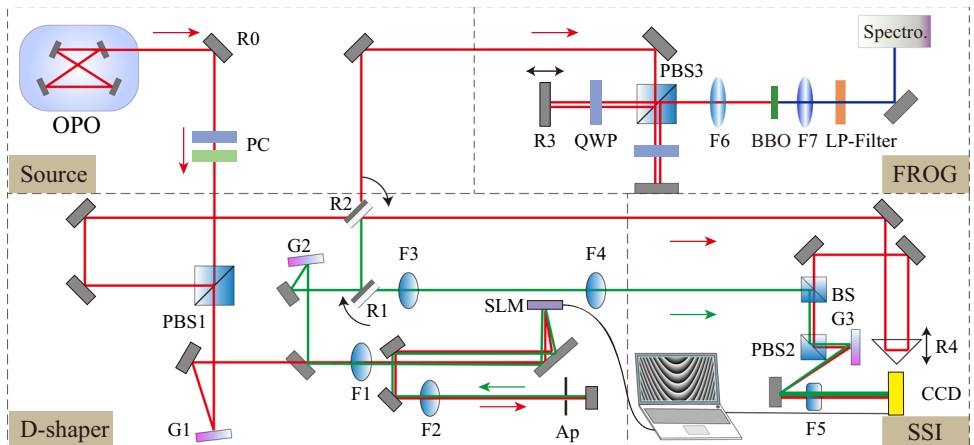

**Fig. 5 | Experimental setup for realizations of fractional Schrödinger equation, with constituents "Source", "D-shaper", "FROG", and "SSI", respectively.** Labels are defined as follows. BS: a beam splitter; PBS1-3: polarization beams splitters; R0: a highly-reflective mirror at 800 nm; R1-2: flip (90˚) mirrors; R3-4: movable mirrors fixed on the translation stage; G1-3: optical gratings with grating densities 1200, 1200, and 600 per mm, respectively; F1-4: convex lenses with the focus length 400, 400, 400, and 300 mm, respectively; F5: a cylinder lens with the focus length 200 mm; F6: a 50 mm lens at 800 nm; F7: a 50 mm lens at 400 nm. BBO: Beta-BaB2O4 nonlinear crystal (type-II); LP-Filter: low-pass filter at 500 nm. SLM: spatial light modulator (HAMAMATSU LCOS-SLM, with the pixels number -1272 × 1024 and pixel pitch -12.5 μm); Ap: Adjustable aperture, OPO compact OPO VIS (Chameleon), PC: polarization controller based on half- and quarter- wave plate, CCD: charge-coupled device camera, Spectro.: spectrometer HR4000CG-UV-NIR (Ocean Optics).

cannot be distinguished, and the "fractional-phase protection" effect will disappear. Based on the results shown in Fig. 4, this overlap is negligible when $\alpha$ is set between 0 and 1. As mentioned above, this effect may find applications in designing robust optical data-transmission protocols.

## Discussion

In this work, we have produced the first experimental realization of the fractional-GVD optical medium, which is modeled by the FSE (fractional Schrödinger equation) in the temporal domain. The experiment makes use of two pulse-shaping holograms, one producing appropriate initial conditions and the other emulating an optical Lévy waveguide. The temporal and spectral fields were reconstructed employing the single-shot SSI (spatial-spectrum interferometry). Several noteworthy dynamical effects have been observed through adjusting parameters of the equivalent Lévy waveguide and applying appropriate phases to the initial pulse. These include the observation of solitary pulses, collision between them, "rain-like" multiple pulses, and double Airy pulse structures. Sub-Fourier patterns, which may simulate sub-Planck structures in quantum mechanics, are produced too using the Wigner-function distribution for the pulses in the time–frequency chronocyclic space. When the third-order spectral phase is applied to the input, the trajectory of the resulting Airy pulses can be efficiently controlled using the LI (Lévy index). Furthermore, the propagation of the pulses carrying the fractional phase reveals the "fractional-phase protection" effect when it passes through the regular medium.

Thus, these findings provide the experimental realization of fractional optical media. The results offer applications to engineering optical pulses and their use in data-transmission schemes. Compared to the previous proposal of implementing FSE in the spatial domain via using an optical resonator[9], our implementation of FSE in the temporal domain has several advantages. Firstly, the current linear setup is feasible as only one-dimensional Fourier optics (compared to two-dimensional required for the spatial domain) needs to be considered, for which the optical cavity is not required. Secondly, we have seen that the pulse dynamics under the FSE not only depend on the LI ($\alpha$), but are also strongly influenced by the initial condition[37,51], which is easier to change by shaping the input pulse in the temporal domain. However, manipulating the initial spatial profile in the cavity's regime may destroy the stability of the cavity. Thirdly, regarding possible extensions of

the work for FSE in the future, such as introducing a potential function, in the current regime based on the temporal mode, this can be engineered directly by employing an electronic optical modulator[22,40,52]. This may be more complex for the cavity's regime, as a spatially varying potential function will also affect the stability of the cavity. Lastly, since no changes to the spatial profile are made and fractional phases are preserved in regular dispersive media, temporal modes can be readily introduced into the current fiber-based communication or waveguide infrastructures. Further, implementing the FSE in the temporal domain makes it natural to include nonlinear materials, which is a promising direction too[12].

It should be noted that the temporal FSE is emulated here by means of two programmable holograms, which is still an indirect implementation of the fractional dispersion. How to create a fractional dispersive material and realize the FSE directly is still an open question[12]. In principle, such a realization may be facilitated in artificial optical materials[37,53]. It is also relevant to mention that the processing time for each pulse measurement is -1 s, including hologram loading, interference-pattern recording, and pulse reconstruction. It is limited mainly by the refresh rate of the pulse shaper, CCD camera, and the computer's processing power. Also, the temporal window $\simeq 8$ ps of the currently used shaper system does not allow us to observe the dynamics on longer distances. These limitations may be relaxed by using an optical grating with a larger number of grooves and the input of the larger beam size.

## Methods

### Experimental setup for the realization of the FSE

Figure 5 shows the experimental setup built for the realization of the FSE. It is divided into four sections: the Source, D-Shaper, SSI and FROG. The Source is based on an OPO system which produces ultrafast pulses (with duration $\simeq 100$ fs at the carrier wavelength 810 nm). They are split into two paths by a polarizing beam splitter (PBS), with a polarization controller used to balance the power between the PBS output ports. One pulse is coupled into the SSI system as a reference. The second, signal pulse, is sent into the pulse shaper (D-shaper) that includes two optical gratings and one SLM (pixels: 1280 × 1024). We split the SLM surface into two sections, so that the pulse is first shone on the top hologram, which is used to set the initial pulse. The pulse is then re-imaged onto the bottom section, where the phase profile is produced by the propagation through a Lévy waveguide or a regular

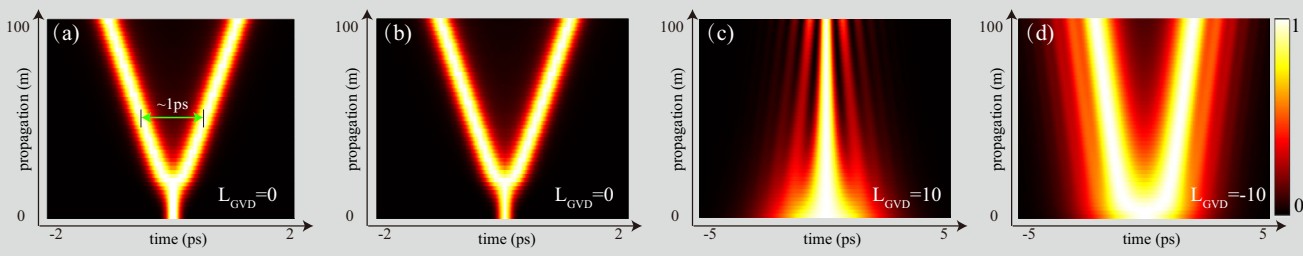

**Fig. 6 | Simulations of the FSE with α = 1. a** The pulse dynamics, as produced by the analytical result (Eq. (18)). **b** The numerical simulation based on the Fourier transform. **c**, **d** The dynamics for $L_{GVD} = 10$ and $-10$ m, respectively ($L_{GVD} = 0$ in cases (**a**) and (**b**)). The parameters are $\beta_2 = -21 \times 10^{-3}$ ps²/s and $D = 21 \times 10^{-3}$ps/s, $\omega_0 = 10$ ps⁻¹. The color-coding in these panels represents the relative intensity, which is normalized so that to uniformly represent values from zero to one.

dispersion material (no fractional component). The two pulse beams finally interfere in the SSI system, allowing the single-shot measurement of the amplitude and phase. The collinear FROG system is used to determine the initial amplitude and phase of the two pulses and to calibrate the SSI. The collinear FROG system deals with pulse duration 317 and 348 fs for the reference and signal pulses, respectively. The bandwidth of these pulses is 6.59 and 4.84 nm, respectively. These measurements are necessary for the calibration of SSI. For the pulse-shaper system, the measured frequency resolution in the Fourier plane of the 4*f* setup (on the SLM surface) is ≃ 1.88 nm/mm, and the temporal window is ≃ 8 ps. More details concerning each section of the setup can be found in Supplementary Notes 1–3.

### The treatment of the FSE

To solve the FSE (Eq. (4)), we consider an input pulse with the second-order spectral phase:

$$\tilde{\Psi}(z=0) = \exp\left(-\frac{\omega^2}{2\omega_0^2} - \frac{i}{2}\beta_2 L_{GVD}\omega^2\right). \tag{15}$$

Here, $\beta_2 L_{GVD}$ represents the second-order GVD, and $\omega_0$ is the bandwidth of the input pulse with the Gaussian distribution in the frequency domain. The FSE produces the corresponding output at propagation length $L$,

$$\tilde{\Psi}(z=L) = \exp\left[-i\left(\frac{D}{2}|\omega|^\alpha\right)L - \frac{\omega^2}{2\omega_0^2} - \frac{i}{2}\beta_2 L_{GVD}\omega^2\right]. \tag{16}$$

Then, the application of the Fourier transform yields

$$\mathcal{F}^{-1}\left[\tilde{\Psi}(z=L)\right]$$
$$= \int_{-\infty}^{+\infty} \exp\left[-i\left(\frac{DL}{2}|\omega|^\alpha + \omega t + \frac{\beta_2}{2}\omega^2 L_{GVD}\right) - \frac{\omega^2}{2\omega_0^2}\right] d\omega$$
$$\equiv \int_{-\infty}^{0} \exp\left[-i\left(-\frac{DL}{2}\omega^\alpha + \omega t + \frac{\beta_2}{2}\omega^2 L_{GVD}\right) - \frac{\omega^2}{2\omega_0^2}\right] d\omega$$
$$+ \int_{0}^{+\infty} \exp\left[-i\left(\frac{DL}{2}\omega^\alpha + \omega t + \frac{\beta_2}{2}\omega^2 L_{GVD}\right) - \frac{\omega^2}{2\omega_0^2}\right] d\omega. \tag{17}$$

Equation (17) admits an analytical solution for $\alpha = 1$, cf. refs. [9,51]:

$$\mathcal{F}^{-1}\left[\tilde{\Psi}(z=L)\right] = \sqrt{\frac{\pi}{a}}\left(\exp\left(-\frac{(DL-2t)^2}{4a}\right)(1-if_1) + \exp\left(-\frac{(DL+2t)^2}{4a}\right)(1-if_2)\right),$$
$$\begin{cases} a = 2i\beta_2 L_{GVD} + 2/\omega_0^2, \\ f_1 = \text{erfi}\left(\frac{DL-2t}{2\sqrt{a}}\right), \\ f_2 = \text{erfi}\left(\frac{DL+2t}{2\sqrt{a}}\right), \\ \text{erfi}(x) \equiv -\frac{2i}{\sqrt{\pi}}\int_0^{ix} \exp(-t^2)dt. \end{cases} \tag{18}$$

In the case of $\beta_2 L_{GVD} = 0$, $a$ is a real number, and solution (18) contains two temporal packets without frequency chirp. The corresponding solutions seem as a superposition of two Gaussians with separation $DL$; hence the separation grows linearly with the increase of transmission length $L$. This feature is observed in Fig. 6a, which is plotted in terms of the analytical expression (18), and agrees with the results of the numerical calculations based on the Fourier transform, which is shown in Fig. 6b. In Fig. 6a, the temporal separation is $DL \approx 1$ ps for $L = 50$ m.

Figure 6c, d display the pulse dynamics based on Eq. (18). It is seen that the initial dispersion, determined by the parameter $L_{GVD}$, strongly affects the emerging pattern, producing the "rain-like" pattern built of many pulses. This happens as the variation of $L_{GVD}$ changes the value of $a$ in Eq. (18), thus affecting the pulse pattern. For fractional LI values, analytical solutions are unavailable, and the numerically implemented Fourier transform should be used.

## Data availability
The experiment data that support the findings of this study are available from the corresponding authors, S.L. (dr.shilongliu@gmail.com) or E.K. (ekarimi@uottawa.ca), upon reasonable request.

## Code availability
The custom code for numerical simulation of the fractional Schrödinger equation and reconstruction of the pulse are available from the corresponding authors, S.L. (dr.shilongliu@gmail.com) or E.K. (ekarimi@uottawa.ca), upon reasonable request.

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

## Acknowledgements

The authors would like to thank Prof. R.W. Boyd for providing the spectrometer used in the FROG system. We appreciate Prof. Alejandro Aceves (University of Arizona) for discussions in the theoretical section; Prof. Denis V. Seletskiy (Polytechnique Montréal) for discussions in the main results. Also, we thank Ende Zuo (University of Ottawa) and Christine Tao (A.U.G. Signals Ltd.) for improvements in figure aesthetics. S.L. acknowledges the support of the International Postdoctoral Exchange Fellowship Program of China Postdoctoral Council (2020020). This work was supported by Canada Research Chairs (CRC), Canada First Research Excellence Fund (CFREF) Program, NRC-uOttawa Joint Centre for Extreme Quantum Photonics (JCEP) via High Throughput and Secure Networks Challenge Program at the National Research Council of Canada, and the Israel Science Foundation via grant No. 1695/22.

## Author contributions

S.L., B.A.M., and E.K. proposed the original idea. E.K. provided full support in the experiments. B.A.M. elaborated on the theoretical part. S.L.

conducted experiments and wrote the initial draft with Y.Z. All authors took part in producing the final manuscript.

## Competing interests

The authors declare no competing interests.
