## [Peer Review File · Nature Communications]

Experimental realisations of the fractional Schrödinger equation in the temporal domainREVIEWER COMMENTS

Reviewer #1 (Remarks to the Author):

The authors of the manuscript NCOMMS-22-25210-T opened up a new field of application for fractional quantum mechanics - laser physics. They developed first experimental realization of a fractional-GVD optical medium described by the fractional Schrodinger equation in the temporal domain.

More precisely, the authors found a physical setting when the quantum wave function receives a fractional phase shift due to the impact of Levy flights with an arbitrary Levy index. The fractional Schrödinger equation is a suitable mathematical tool for studying quantum phenomena under these conditions. The theoretical consideration is supported by numerical simulations, which are closely resembled by to the experimental results.

The key discovery is the experimental implementation of the fractional Schrödinger equation for femtosecond laser pulses in the time domain. The theoretical consideration is supported by numerical simulations, the results of which are very similar to the experimental observations.

The manuscript is well written and meets the high requirements of Nature Communications for scientific subject matter and presentation style.

The manuscript NCOMMS-22-25210-T will be of great interest to researchers and graduate students looking for new theoretical methods and interesting applications in the fields of laser physics, optical signal processing, and quantum communication system design.

I recommend the manuscript NCOMMS-22-25210-T for publication.

Reviewer #2 (Remarks to the Author):

The manuscript by Liu et al. presents an experimental realization of space fractional quantum mechanics in an all-optical setup. The manuscript is well written and understandable, and the experimental findings will have significant impact on the field. I therefore recommend publication of the manuscript in Nat. Commun. after the authors considered the following minor comments.

A bit more background information on the physical implications of space-fractional quantum mechanics might help the reader to appreciate the relevance and physics of the manuscript. For instance,

- When the authors introduce and explain the fractional Schrödinger equation (1). Please add to the main text a physical explanation how this effective dynamics is obtained. For instance, what determines physically the magnitude of the coefficients D and β_k ?
- The definition of the fractional time derivative (2) includes an integral over all times. How can this be realised by a real material/setup, whose linear response always respects causality (the Kramers-Kronig relation)? Where can one see the associated dissipation/damping?
- L_{GVD} is introduced as the 'length of the shaping segment' however later the authors realise $L_{\text{GVD}} < 0$. Please clarify.
- Is the 'beam-splitting' visible Fig 2 Q1 – Q4 coherent? Specifically, can the two branches be recombined to show interference?
- I do not understand Fig 4. How exactly are the weights defined? What is the solid line?

Reviewer #3 (Remarks to the Author):

The fractional Schrödinger equation (FSE) is a hot topic and has attracted an increasing interest recently due to universal and general form. But the challenge of its experimental realization is always existing. Especially, what kind of medium can support the model of FSE. In this manuscript, the authors claimed that they have realized the FSE in time domain experimentally. It seems that the presentation is not easy to follow and these are some questions should be addressed clearly.

(1) Equation (2) contains the fractional dispersion, the normal dispersion and the effective potential. This is not pure FSE. Based on the equation, the question is what kind of media for pulse propagation? Did the authors use normal optical fiber? If so, then the pre-chirp was imposed on the input pulse for canceling the effect of fiber dispersion? I recommended the author present a clear description.

(2) What is Levy waveguide? The optical Lévy waveguide is realized directly or indirectly? To me, the FSE is demonstrated still by using Fourier transform, not find a real physical system that can be described directly by FSE. I think the authors should maybe make this point clear. In previous literature, some schemes are proposed to see fractional diffraction effect by using some lens and phase masks, can the authors make a comparison of their scheme with previous schemes? What is the advantage of the scheme in this manuscript?

(3) Equation (18) was reported previously in [11] and [Mathematics, Vol. 4, Iss. 2, p. 31 (2016)].

(4) The authors emphasize their demonstration in temporal domain in the title. I wonder why the temporal domain is important.

(5) As to the “fractional-phase protection” in this manuscript, it seems that the input beam should carry fractional phase and it reserves even the input beam passes through a regular medium. This is really nontrivial, but it is hard to understand why the normal dispersion does not affect its property. Can the authors show more explanations?

Reviewer 1

The authors of the manuscript NCOMMS-22-25210-T opened up a new field of application for fractional quantum mechanics - laser physics. They developed first experimental realization of a fractional-GVD optical medium described by the fractional Schrodinger equation in the temporal domain.

More precisely, the authors found a physical setting when the quantum wave function receives a fractional phase shift due to the impact of Levy flights with an arbitrary Levy index. The fractional Schrödinger equation is a suitable mathematical tool for studying quantum phenomena under these conditions. The theoretical consideration is supported by numerical simulations, which are closely resembled by to the experimental results.

The key discovery is the experimental implementation of the fractional Schrödinger equation for femtosecond laser pulses in the time domain. The theoretical consideration is supported by numerical simulations, the results of which are very similar to the experimental observations.

The manuscript is well written and meets the high requirements of Nature Communications for scientific subject matter and presentation style.

The manuscript NCOMMS-22-25210-T will be of great interest to researchers and graduate students looking for new theoretical methods and interesting applications in the fields of laser physics, optical signal processing, and quantum communication system design.

I recommend the manuscript NCOMMS-22-25210-T for publication.

Reply: The authors would like to express their sincere thanks to the Reviewer for reading the manuscript, providing feedback, and their recommendation.

Reviewer 2

The manuscript by Liu et al. presents an experimental realization of space fractional quantum mechanics in an all-optical setup. The manuscript is well written and understandable, and the experimental findings will have significant impact on the field. I therefore recommend publication of the manuscript in Nat. Commun. after the authors considered the following minor comments.

Reply: We would like to thank the Reviewer for reading the manuscript and for giving a very positive evaluation of this work. We have addressed all the comments, with the respective modifications in the manuscript highlighted in blue.

- A bit more background information on the physical implications of space-fractional quantum mechanics might help the reader to appreciate the relevance and physics of the manuscript. For instance, When the authors introduce and explain the fractional Schrödinger equation (1). Please add to the main text a physical explanation how this effective dynamics is obtained. For instance, what determines physically the magnitude of the coefficients D and β_k ?

Reply: In order to clarify the physical purport of FSE more clearly, we have added the following explanation in the main text (below Eq. (2)):

On the right-hand side of Eq. (2), the first term is the fractional time derivative with the $LI \alpha$, where D is the fractional dispersion coefficient. The second term represents the integer derivative that corresponds to the k -th regular GVD with the coefficient β_k , and the last term is the potential $V(\tau)$. In optics, such a complex dispersion material involving both these ingredients may be realized by means of an artificial photonic structure [37]. In that case, the model defined in Eq. (2) may also include dissipation, i.e., dispersive losses, which are not considered in this work.

- The definition of the fractional time derivative (2) includes an integral over all times. How can this be realised by a real material/setup, whose linear response always respects causality (the Kramers-Kronig relation)? Where can one see the associated dissipation/damping?

Reply: It is not easy to realize the fractional time derivative in Eq. (3) directly in the time domain. Therefore, we engineer it indirectly in the frequency domain through Eq. (4). Based on Eq. (4), the causality relation between the fractional spectral phase and the fractional time derivative always holds in the current linear system. Regarding the dissipation/damping, i.e., the frequency-dependent dispersion losses, they would definitely affect the dynamics. However, they are much smaller in our designed shaper system, because the hologram carries only the phase information instead of the amplitude that represents the dissipation. Therefore, we do not include them in the theoretical analysis. These aspects of the model are clarified in the added description under Eq. (2).

- L_{GVD} is introduced as the ‘length of the shaping segment’ however later the authors realise $L_{\text{GVD}} < 0$. Please clarify.

Reply: L_{GVD} together with β_k is used to produce the corresponding phase shift $\phi_{\text{GVD}} = (\beta_k \times L_{\text{GVD}})$. For the definiteness’s sake, we set β_k to be a constant, i.e., $\beta_2 = -21 \times 10^{-3} \text{ ps}^2/\text{m}$, and only discuss effects of L_{GVD} . We have clarified this in the text below Eq. (5) as follows.

For the spectral phase ϕ_{GVD} , it can be positive or negative depending on the sign of β_2 . For the definiteness’s sake, we set β_k to be a constant, i.e., $\beta_2 = -21 \times 10^{-3} \text{ ps}^2/\text{m}$, and only discuss effects of L_{GVD} .

- Is the ‘beam-splitting’ visible Fig 2 Q1 – Q4 coherent? Specifically, can the two branches be recombined to show interference?

Reply: Here, the pulse beam-splitting originates from the same one pulse. Therefore, yes, it is coherent due to the stable relative temporal phase. The interference will appear if the relative delay is made close to zero.

- I do not understand Fig 4. How exactly are the weights defined? What is the solid line?

Reply: In the revised manuscript, we have updated Fig. 4 to make it clearer. Here, the weights W_j (in the first line of Eq. (14)) represent the reconstructed spectral phase for $\alpha = 0.4, 0.6, 0.8,$ and $1,$ respectively. We note that we have not included this weighting term W_j in Eq. (14), in which we have now added. In Fig. 4, the red balls illustrate how both the weighting (W_j) and $LI(\alpha)$ of the fractional phase stay mostly unchanged when passing through a regular dispersion material; while the yellow balls represent how the weighting of the second-order spectral phase ($W_{k=2}$) given in the second line of Eq. (14) change during transmission. The solid line, connecting the red or yellow balls, is used to show their relative distribution. These details are now clarified in the revised caption of Fig. 4 as follows:

The four bottom panels show the weightings and LI for the reconstructed phase structure given by Eq. (14) for propagation distances $z = 1 m, 3 m, 6 m,$ and $9 m,$ respectively. In these panels, the four red balls represent four different fractional-phase terms in the first line of Eq. (14), which are generated by input-phase terms with parameters $(\alpha_j, W_j) = (0.4, 0.73), (0.6, 0.07), (0.8, 0.51),$ and $(1.0, 0.22),$ respectively. The horizontal axis in each panel represents the set of values of $LI(\alpha_j) = 0.4, 0.6, 0.8, 1$ and $2,$ respectively; the vertical axis represents values of the relative weight W_j from 0 to 0.7. Yellow balls denote the relative weights of the second-order phase term $W_{k=2}$. The solid line, connecting the red or yellow ball, is used to show their relative distribution.

Reviewer 3

The fractional Schrödinger equation (FSE) is a hot topic and has attracted an increasing interest recently due to universal and general form. But the challenge of its experimental realization is always existing. Especially, what kind of medium can support the model of FSE. In this manuscript, the authors claimed that they have realized the FSE in time domain experimentally. It seems that the presentation is not easy to follow and these are some questions should be addressed clearly.

Reply: We would like to thank the Reviewer for reading our manuscript and for sharing their valuable comments. We agree that experimental realisation of the fractional Schrödinger equation (FSE) is still a challenge, and it is a relevant topic to find a physical medium to realise FSE. We have addressed all the reviewer's comments to improve the clarity and readability of the manuscript, as detailed below. The respective changes in the revised text are marked in red. We believe the updated version is easier to read.

(1) Equation (2) contains the fractional dispersion, the normal dispersion and the effective potential. This is not pure FSE. Based on the equation, the question is what kind of media for pulse propagation? Did the authors use normal optical fiber? If so, then the pre-chirp was imposed on the input pulse for canceling the effect of fiber dispersion? I recommended the author present a clear description.

Reply: Yes, Eq. (2) is not a pure FSE, as it includes not only the fractional time derivative but also the regular dispersion. When we ignore the regular dispersion, it should be a pure FSE. One of the reasons which determine the form of the model is to make it more comprehensive. The other is that we can address the combined effects of both the fractional and regular dispersion in some of our results, i.e., in the section on 'fractional-phase protection'. The creation of artificial optical materials combining these features is quite plausible. Following Reviewer 2 having a similar question regarding Eq. (2), we have added the following clarifications in the blue text under Eq. (2) of the main text:

On the right-hand side of Eq. (2), the first term is the fractional time derivative with the $LI \alpha$, where D is the fractional dispersion coefficient. The second term represents the integer derivative that corresponds to the k -th regular GVD with the coefficient β_k , and the last term is the potential $V(\tau)$. In optics, such a complex dispersion material involving both these ingredients may be realized by means of an artificial photonic structure [37]. In that case, the model defined in Eq. (2) may also include dissipation, i.e., dispersive losses, which are not considered in this work.

Regarding the pre-chirp in the input pulse, it is a relevant point. We don't use any fibers in the experiment, while some fused silica elements are employed, such as the beam splitters in setup, so some pre-chirp does appear in the input pulse. This is why the minimum position for the duration of the propagating pulse (Fig. 2(b)-B2) is not at the initial zero point. The pre-chirp can also be verified in Fig. 1(a) and (b) in the supplementary material. We have added a clarification of this point in the revised text in red at the end of page 4.

Here, the minimum of the pulse duration is shifted slightly from the expected zero point, which implies a minor difference in the frequency chirp between the reference and signal pulse. This additional chirp mainly comes from the various fused silica optical elements, which could be compensated by adding a small opposite spectral phase on the hologram (see details in Section 2 of the Supplementary Material).

(2) What is Levy waveguide? The optical Lévy waveguide is realized directly or indirectly? To me, the FSE is demonstrated still by using Fourier transform, not find a real physical system that can be described directly by FSE. I think the authors should maybe make this point clear. In previous literature, some schemes are proposed to see fractional diffraction effect by using some lens and phase masks, can the authors make a comparison of their scheme with previous schemes? What is the advantage of the scheme in this manuscript?

Reply: The Levy waveguide is a general concept for a material that could produce an element producing the effective fractionality. Indeed, the current work realises the FSE in an indirect manner with the help of the Fourier transform, given by Eq. (4). To our best knowledge, there is currently no feasible physical platform to realise FSE in a direct manner. Although the manner is indirect, the dynamics governed by FSE are accurately demonstrated with the effects seen in a real physical wave packet. We clarify these points in the section of the discussion. In previous literature (Ref [9]), Fourier optics was also employed to numerically simulate the FSE effects of the spatial beam inside a cavity. Here, we chose the spectral-temporal domain due to several advantages via comparing with the previous proposal:

- The temporal FSE, is more feasible to be experimentally realised because one only needs to consider one-dimensional Fourier optics. For the spatial case proposed in [9], the experimental realisation becomes much more complicated. An optical resonator cavity needs to be constructed and stabilised; it also requires additional gains to compensate for the losses within the cavity.
- From our results, it could be observed that the fractional pulse dynamics not only depend on the $LI(\alpha)$, but is also strongly influenced by the initial condition (the profile of the initial pulse). This could be an issue in the spatial regime using the proposed cavity setup, as small changes in the initial beam profile may destroy the stability of the cavity.
- The other advantage of our scheme over the spatial cavity setup is in introducing a potential function V . For our scheme of spectral-temporal modes; it is possible to directly load various potential $V(\tau)$ by using a high-speed electronic optical modulator. The spatial cavity design may have stability issues when spatially changing the potential function $V(x, y)$.
- Finally, the transverse spatial structure of the temporal mode pulse is still a Gaussian profile, which can be implemented directly into the fibre-based communication infrastructure.

In general, working in the spectral-temporal domain is much more versatile than in the spatial domain. The following has been added to the Discussion section to reflect the above:

Compared to the previous proposal of implementing FSE in the spatial domain via using an optical resonator [9], our implementation of FSE in the temporal domain has several advantages. Firstly, the current linear setup is feasible as only one-dimensional Fourier optics (compared to two-dimensional for spatial) need to be considered, in which the optical cavities is not required. Secondly, we have seen that the pulse dynamics under the FSE, not only depend on the $LI(\alpha)$, but is also strongly influenced by the initial condition [37,51]; for which the initial is easier to change by shaping the input pulse in the temporal domain. However, manipulating of the initial spatial profile in the cavity's regime may destroy the stability of the cavity. Thirdly, regarding possible extensions work for FSE in the future, such as, introducing a potential function, with the current regime with temporal mode, this can be engineered directly by employing an electronic optical modulator [22,40,52]. While this may be more complex for the cavity's regime, as a spatially varying potential function will also affect the stability of the cavity. Lastly, since no changes to the spatial profile are made and fractional phases are preserved through regular dispersive media, temporal modes can be readily implemented into the current fiber-based communication or waveguide infrastructure.

(3) Equation (18) was reported previously in [11] and [Mathematics, Vol. 4, Iss. 2, p. 31 (2016)].

Reply: We have now cited this work as Ref [51] in the revised manuscript.

(4) The authors emphasize their demonstration in temporal domain in the title. I wonder why the temporal domain is important.

Reply: Just as the reply for the second comment, the temporal mode offers many advantages in optical information processing. As for the title, since all the dynamics are investigated in the temporal domain, therefore we think it's relevant to emphasise this in the title.

(5) As to the “fractional-phase protection” in this manuscript, it seems that the input beam should carry fractional phase and it reserves even the input beam passes through a regular medium. This is really nontrivial, but it is hard to understand why the normal dispersion does not affect its property. Can the authors show more explanations?

Reply: From Eq. (14), it can be seen that a regular dispersion material will contribute an extra phase factor (phase term in the second expression of Eq. (14)) which does not affect the fractional phase on the initial pulse (phase term in the first expression of Eq. (14)), thus resulting in the ‘fractional-phase protection’. However, this does not always hold, which depends on the value of $LI(\alpha)$. As α becomes closer to 2, the similarity will become more higher between the fractional and non-fractional case of $k = 2$. When $\alpha = 2$, the two types of phases cannot be distinguished, and the ‘fractional-phase protection’ effect will disappear. In our experimental results, seen in Fig. 4 and Movie 3, this overlap is negligible when α is set between 0 and 1. We have added some explanation of this feature at the end of Page 6 as follows:

It should be noted the “fractional-phase protection” effect may not hold for all values of $LI \alpha$. With α becoming closer to 2, the profile of the fractional spectral phase will become more similar to the non-fractional case of $k = 2$. When $\alpha = 2$, the two types of phases cannot be distinguished, and the ‘fractional-phase protection’ effect will disappear. Based on the results in Fig. 4, this overlap is negligible when α is set between 0 and 1.

REVIEWER COMMENTS

Reviewer #2 (Remarks to the Author):

I thank the authors for their comprehensive reply. They answered all my questions. I recommend publication of the manuscript in its current form.

Reviewer #3 (Remarks to the Author):

In the revised manuscript, the authors replied the comments carefully. Even though the authors claim that they realized fractional Schroedinger equation, it is still not so easy to follow the idea of the work. One deterministic thing is that this work is in an indirect way, which is also inquired by other referees. However, the authors treat it in a vague way that misleads researchers to some extent. No doubt, endeavors are still highly desired to realize the fractional Schroedinger equation in a direct way. As a result, the authors are expected to add related statements in the manuscript to clarify the point, for example in the conclusion part, the authors may add that the realization is still in an indirect way, and it is still an open project for realizing the fractional Schroedinger equation in a direct manner.

A point-by-point response to the reviewers' comments

Reviewer #2 (Remarks to the Author):

I thank the authors for their comprehensive reply. They answered all my questions. I recommend publication of the manuscript in its current form.

Reply: We thank the Reviewer for the recommendation.

Reviewer #3 (Remarks to the Author):

In the revised manuscript, the authors replied the comments carefully. Even though the authors claim that they realized fractional Schroedinger equation, it is still not so easy to follow the idea of the work. One deterministic thing is that this work is in an indirect way, which is also inquired by other referees. However, the authors treat it in a vague way that misleads researchers to some extent. No doubt, endeavors are still highly desired to realize the fractional Schroedinger equation in a direct way. As a result, the authors are expected to add related statements in the manuscript to clarify the point, for example in the conclusion part, the authors may add that the realization is still in an indirect way, and it is still an open project for realizing the fractional Schroedinger equation in a direct manner.

Reply: We thank the Reviewer for the relevant consideration. We agree with the comment that the setup elaborated in this work realises the fractional dispersion indirectly. Nevertheless, the dynamics of the output laser pulse is correctly modelled by the temporal fractional Schrödinger equation (FSE); hence the present setup provides an appropriate realisation of the FSE. It is true that the direct realisation of the fractional dispersion or diffraction remains a relevant topic. To address this point, we have added the following text at the end of the Discussion section, singled out by the red font.

It should be noted that the temporal FSE is emulated here by means of two programmable holograms, which is still an indirect implementation of fractional dispersion. How to create a fractional dispersive material and realize the FSE directly is still an open question. In principle, such a realization may be facilitated in artificial optical materials [37, 53].